# Differences in the inflammatory proteome of East African and Western European adults and associations with environmental and dietary factors

Godfrey S Temba[1,2], Nadira Vadaq[1], Vesla Kullaya[2,3], Tal Pecht[4,5], Paolo Lionetti[6], Duccio Cavalieri[7], Joachim L Schultze[4,5,8], Reginald Kavishe[2], Leo AB Joosten[1,9], Andre J van der Ven[1], Blandina T Mmbaga[3,10], Mihai G Netea[1,11], Quirijn de Mast[1]*

[1]Department of Internal Medicine, Radboudumc Center for Infectious Diseases, Radboudumc Research Institute for Medical innovation (RIMI), Radboud University Medical Center, Nijmegen, Netherlands; [2]Department of Medical Biochemistry and Molecular Biology, Kilimanjaro Christian Medical University College, Moshi, United Republic of Tanzania; [3]Kilimanjaro Clinical Research Institute, Kilimanjaro Christian Medical Center, Moshi, United Republic of Tanzania; [4]Department for Genomics and Immunoregulation, Life & Medical Sciences (LIMES) Institute, University of Bonn, Bonn, Germany; [5]Systems Medicine, German Center for Neurodegenerative Diseases (DZNE), Bonn, Germany; [6]Departement NEUROFARBA, University of Florence – Gastroenterology and Nutrition Unit, Meyer Children's Hospital, Florence, Italy; [7]Department of Biology, University of Florence, Florence, Italy; [8]PRECISE Platform for Single Cell Genomics and Epigenomics, German Center for Neurodegenerative Diseases (DZNE) and University of Bonn, Bonn, Germany; [9]Department of Medical Genetics, Iuliu Hatieganu University of Medicine and Pharmacy, Cluj-Napoca, Romania; [10]Department of Paediatrics, Kilimanjaro Christian Medical University College, Moshi, United Republic of Tanzania; [11]Department of Immunology and Metabolism, Life & Medical Sciences (LIMES) Institute, University of Bonn, Bonn, Germany

*For correspondence: quirijn.demast@radboudumc.nl

Competing interest: The authors declare that no competing interests exist.

**Abstract** Non-communicable diseases (NCDs) are rising rapidly in urbanizing populations in sub-Saharan Africa. Assessment of inflammatory and metabolic characteristics of a urbanizing African population and the comparison with populations outside Africa could provide insight in the pathophysiology of the rapidly increasing epidemic of NCDs, including the role of environmental and dietary changes. Using a proteomic plasma profiling approach comprising 92 inflammation-related molecules, we examined differences in the inflammatory proteome in healthy Tanzanian and healthy Dutch adults. We show that healthy Tanzanians display a pro-inflammatory phenotype compared to Dutch subjects, with enhanced activity of the Wnt/β-catenin signalling pathway and higher concentrations of different metabolic regulators such as 4E-BP1 and fibroblast growth factor 21. Among the Tanzanian volunteers, food-derived metabolites were identified as an important driver of variation in inflammation-related molecules, emphasizing the potential importance of lifestyle changes. These findings endorse the importance of the current dietary transition and the inclusion of underrepresented populations in systems immunology studies.

## Editor's evaluation

The manuscript by Temba and colleagues describe an essential aspect of human research, i.e., the variability of different populations which have different genetic backgrounds and are exposed to different diets and environments. As most research is carried out in developed countries, we need to understand and generate background data in other populations, such as the described here in Tanzania. The fundamental findings refer to enhanced pro-inflammatory phenotype in plasma of individuals from Tanzania as compared to individuals from Holland. Food-derived metabolites were identified as a driver of variation in inflammation-related molecule expression. Whether pro-inflammatory phenotypes associate with changes in life span or risk of chronic disease and whether changes in life style will reverse this pro-inflammatory phenotype and related outcomes clearly deserve further investigation in the future.

## Introduction

The human immune response is tightly regulated by a complex and intricate network of pro-and anti-inflammatory cytokines, chemokines and other immuno-metabolic mediators. Prolonged dysregulation of this network may result in an unresolved pro-inflammatory state, which is central to the development of a wide range of non-communicable diseases (NCDs), including cardiovascular disease, diabetes, rheumatic and inflammatory bowel diseases, and even malignancies (*Choi et al., 2015*; *Moore and Tabas, 2011*; *Seyedsadjadi and Grant, 2021*). The inflammatory response is highly variable among healthy individuals as a consequence of genetic and non-genetic factors (*Brodin and Davis, 2017*). These include intrinsic factors such as age and sex, as well as environmental exposures such as diet and past infections (*Liston et al., 2021*). Not surprisingly, variation in inflammatory proteins between populations has been reported (*Schutte et al., 2012*; *Schutte et al., 2006*). Understanding the nature of this variation and the factors involved is key for understanding the dynamics of infectious as well as immune-mediated pathology across populations.

Worldwide, many communities are currently undergoing a rapid urbanization process with a transition of lifestyle, and diet, characterized by a more sedentary lifestyle and a shift from traditional high-fiber diets to a diet richer in processed foods, animal fat and simple carbohydrates. Also, environmental exposures change from close contact with animals and smoke from burning wood to petrol gasses. This is accompanied by an epidemiologic transition in which the burden of disease shifts from infectious diseases to NCDs (*Beaglehole et al., 2011*; *Unwin et al., 2010*). This epidemiologic transition is at least partly mediated through effects on the immune system of individuals across communities. We recently reported that urban-living Tanzanians display a pro-inflammatory gene signature and higher ex-vivo cytokine responses compared to rural-living individuals (*Temba et al., 2021*) and that food-derived metabolites are an important driver of this difference. In communities in sub-Saharan Africa, this effect may also be more pronounced than in other populations, as the historically high burden of infectious diseases may have resulted in the selection of genotypes favoring a robust immune response (*Karlsson et al., 2014*). Indeed, we recently showed important differences in the genetic regulation of cytokine responses between healthy Tanzanian and Dutch individuals, with enrichment of interferon pathways in the Tanzanians (*Boahen et al., 2022*). In a context with a declining burden of infectious diseases, alongside a shift to an unhealthy Western-type lifestyle, such a heritable pro-inflammatory phenotype may particularly drive a health-to-disease transition with the onset of NCDs (*Bickler et al., 2018*).

Studies on non-Western populations outside historically wealthy countries are underrepresented in systems-immunology literature, despite the fact that these populations are representative of the majority of the world's population. In addition, these populations offer unique opportunities to increase our understanding of the pathophysiology of NCDs, including the role of diet and environmental exposures. Our present knowledge of the regulation of the immune system in individuals in sub-Saharan Africa and how this compares to individuals in the industrialized world is limited. We investigated the hypothesis that healthy individuals in East Africa have a pro-inflammatory phenotype in comparison to individuals from North-western Europe and that these differences are partly driven by common environmental factors, including diet. Using a 92-plex proteomic panel based on a proximity extension assay technology, we compared the inflammatory proteome of healthy Tanzanians of African origin residing in Tanzania with that of healthy individuals of Western-European ancestry living

in the Netherlands. Next, we studied associations between the proteome with intrinsic factors and environmental exposures, with special emphasis on the food-derived metabolites.

# Results

## Demographics

Data from plasma samples of 318 Tanzanian and 416 Dutch individuals were included in this study. Characteristics of study participants are summarized in *Table 1*, *Figure 1—figure supplement 1*. The Tanzanians had a significantly higher median (IQR) age (30.2 years; 23.4–39.9) than the Dutch (23.0 years; 21.0–26.0; p-value <0.0001) (*Figure 1—figure supplement 1A*). Tanzanian females also had a higher BMI than Dutch females (25.7; 22.6–29.9 vs. 21.5; 20.4–23.1; p-value <0.0001) (*Figure 1—figure supplement 1B*).

## Differences in inflammatory proteome between Tanzanians and Dutch

The inflammatory proteome was measured simultaneously in samples from the Tanzanian and Dutch participants using the Olink 'inflammation' panel, which targets 92 cytokines, chemokines and other inflammation and metabolism-related proteins (Olink Proteomics AB, Uppsala, Sweden). Relative protein concentrations are reported as normalized protein expression (NPX) units, which are on a Log2 scale. Eighteen proteins were excluded from further analysis because their value was below the lower limit of detection in more than 25% of samples in both cohorts (*Figure 1—figure supplement 2*). Principal component analysis (PCA) of the remaining 74 proteins revealed a clear separation between Tanzanian and Dutch samples (*Figure 1A*). A volcano plot of differentially expressed proteins (*Figure 1B*) showed that 35 (47%) proteins were significantly higher in the Tanzanians and 20 (27%) lower at an FDR p<0.05, with correction for age and sex and BMI. The most prominently (fold change (FC)) upregulated proteins were two regulators of metabolism: the mTOR substrate and translational repressor 4E-BP1 ($\log_2$ FC 1.9; FDR p=$1.3 \times 10^{-60}$) and FGF21 (fibroblast growth factor 21; $\log_2$ FC 1.3; p=$1.3 \times 10^{-30}$), a hormone produced by the liver that functions as a major regulator of glucose and lipid homeostasis. Obesity and excess carbohydrate and/or insufficient protein intake were reported to increase FGF21 concentrations (*Hill et al., 2018*). Other prominently upregulated proteins in the Tanzanians were interleukin (IL)–17 A ($\log_2$ FC 0.7; p=$9.6 \times 10^{-34}$) and IL-17C ($\log_2$ FC 0.7; p=$1.1 \times 10^{-74}$), and the CC-chemokine family members CCL11/eotaxin (eosinophil chemoattractant; $\log_2$ FC 0.7; p=$7.3 \times 10^{-63}$), CCL3/MIP-1α (macrophage inflammatory protein-1α; $\log_2$ FC 0.6; p=$5.6 \times 10^{-10}$), CCL7/MCP3 (monocyte chemotactic protein 3; $\log_2$ FC 0.6; p=$7.4 \times 10^{-42}$) and CCL8/MCP2 ($\log_2$ FC 0.5; p=$8.4 \times 10^{-23}$). The cytokines Tumour Necrosis Factor (TNF), IL-6, IL-10, and IL-18, as well as oncostatin-M (OSM) and adenosine deaminase (ADA) were also significantly higher in the Tanzanians.

The most prominently downregulated proteins in Tanzanians were ST1A1 (sulfotransferase 1A1; $\log_2$ FC –0.8; p=$2.2 \times 10^{-16}$) and AXIN1 (axis inhibition protein 1; $\log_2$ FC –0.7; p=$6.5 \times 10^{-18}$). ST1A1 is a cytosolic sulfotransferase that catalyzes the sulfonation of endogenous and exogenous compounds (*Wang et al., 2016*). AXIN1 is a negative regulator of the Wnt/β-catenin signaling pathway (*Kikuchi, 1999*). This pathway is increasingly recognized to play an important role in inflammatory diseases, diabetes and cancer (*Jridi et al., 2020*; *Das et al., 2021*). Conversely, CDCP1 (CUB domain-containing protein 1), a transmembrane receptor that is a Wnt signaling promoter (*He et al., 2020*) was significantly up-regulated ($\log_2$ FC 0.7; p=$1.8 \times 10^{-61}$) in the Tanzanian cohort, suggesting enhanced activity of the Wnt/β-catenin signaling pathway in the Tanzanian participants. Finally, Tanzanians had lower levels of the CXC chemokine family members CXCL1, CXCL5, CXCL6, and CXCL8 (IL8). These chemokines mediate among others neutrophil trafficking (*Palomino and Marti, 2015*).

The Olink platform used in this study does not contain adipocytokines and provides relative, rather than absolute, protein concentrations. Therefore, we measured absolute concentrations of a selection of cytokines using an ELLA microfluidics platform, and concentrations of adipocytokines by ELISA. The data have been reported previously (*Temba et al., 2022*), and confirm that Tanzanian participants have significantly higher plasma concentrations of IL-1 receptor antagonist (IL-1Ra), IL-6, and IL-18, together with significantly higher leptin and lower adiponectin concentrations than Dutch participants after correcting for age, sex, and BMI (*Figure 1C*). In addition, Tanzanian females regardless of BMI had higher plasma concentrations of leptin compared to Dutch females (*Figure 1—figure*

**Table 1.** Descriptive characteristics of study participants.

| | Tanzanians | Dutch | p-value |
|---|---|---|---|
| Number | 318 | 416 | |
| Sex, females | 163 (51.3) | 214 (51.4) | ns |
| Age, years | 30 (23–40) | 23 (21–26) | <0.0001 |
| **Age category** | | | <0.0001 |
| 18–30 years | 155 (48.7) | 351 (85.8) | |
| 31–40 years | 84 (26.4) | 15 (3.7) | |
| 41–50 years | 50 (15.7) | 4 (1.0) | |
| 50–60 years | 25 (7.9) | 13 (3.2) | |
| ≥60 years | 4 (1.3) | 26 (6.4) | |
| **BMI** | 23.8 (21.4–27.3) | 22.3 (20.7–24.3) | <0.0001 |
| BMI category | | | <0.0001 |
| ≤24.9 | 194 (61.0) | 337 (84.0) | |
| ≥25–29.9 | 76 (23.9) | 57 (14.2) | |
| ≥30 | 48 (15.1) | 7 (1.7) | |
| **BMI by sex** | | | |
| Male | 22.8 (20.8–24.9) | 23 (21.7–24.6) | ns |
| Female | 25.7 (22.6–29.9) | 21.5 (20.4–23.1) | <0.0001 |
| Smoking (N, %) | 50 (15.7) | 57 (13.7) | ns |
| | | | |
| Study characteristics only relevant for the Tanzanian cohort | | | |
| Residency (N, %) | | | |
| Urban | 250 (78.6) | | |
| Rural | 68 (21.4) | | |
| **Highest level of education (N, %)** | | | |
| Primary | 119 (37.4) | | |
| Secondary | 69 (21.7) | | |
| College | 61 (19.2) | | |
| University | 25 (7.9) | | |
| In training | 44 (13.8) | | |
| **Occupational status (N, %)** | | | |
| Student | 57 (17.9) | | |
| Employed with qualification | 66 (20.8) | | |
| Service or shop sales worker | 133 (41.8) | | |
| Elementary occupation | 62 (19.5) | | |
| **Toilet facility (N, %)** | | | |
| Pit latrine | 75 (23.6) | | |
| Water closet | 243 (76.4) | | |
| **Cooking fuel (N, %)\*** | | | |
| Smoky | 113 (35.5) | | |

*Table 1 continued on next page*

*Table 1 continued*

|  | Tanzanians | Dutch | p-value |
|---|---|---|---|
| Non-smoky | 205 (64.5) |  |  |
| Exposure to animals, yes (N, %) | 138 (43.4) |  |  |
| The course of antibiotics in the past year (N, %) |  |  |  |
| 1–3 courses | 164 (50.8) |  |  |
| >3 courses | 28 (8.7) |  |  |
| None | 131 (40.6) |  |  |
| Last time antibiotic use (N, %) |  |  |  |
| 3–6 months ago | 44 (13.6) |  |  |
| 6–12 months ago | 65 (20.1) |  |  |
| 12 months ago, | 83 (25.7) |  |  |
| None | 131 (40.6) |  |  |
| Access to clean water (N, %) |  |  |  |
| Tap water | 314 (97.2) |  |  |
| Well, canal or river water | 9 (2.8) |  |  |

Comparison between characteristics of the Tanzanian and the Dutch participants was done using Chi-square and, Mann-Whitney U tests; for categorical and continuous variables respectively.
*categories of cooking fuel include firewood, charcoal or kerosene (smoky), or gas and electrical (non-smoky).

*supplement 3*) in line with the previous findings (*Abbas et al., 2004*). There were no significant differences in plasma concentrations of resistin and alpha-1 antitrypsin (AAT).

In a previous study in the same cohort, we identified different genetic loci that were associated with whole-blood cytokine responses to a variety of microbial and synthetic ligands (*Boahen et al., 2022*). We assessed whether these genetic variants were also associated with plasma protein levels. We utilized the top six independent SNP-cytokine response loci and performed matrix pQTL analysis. Different associations were identified, but none reached genome-wide ($p=5 \times 10^{-8}$) or suggestive ($p=5 \times 10^{-6}$) significance (*Supplementary file 1*; P-value unadjusted).

## Associations between inflammation-related proteins and intrinsic and environmental factors

Next, we investigated associations between the inflammation-related proteins with host intrinsic factors such as age and sex, BMI and environmental exposures relevant to the Tanzanian setting. The analyses were variously corrected for age, sex and BMI to assess the impact of one specific factor. In high-income countries, age is a potent driver of immune variation with a shift toward a pro-inflammatory state (*Brüünsgaard and Pedersen, 2003*, *Ferrucci et al., 2005*). In the Dutch participants, advancing age was indeed associated with an increase in inflammation-related proteins, including inflammatory cytokines (IL-6, IL-7, IL-18), IL-15RA, monocyte chemoattractant proteins (MCP2-4), matrix metalloprotease1 (MMP1), the chemokine IL-8 and hepatocyte growth factor (HGF). In contrast, these significant associations were largely absent in the Tanzanians (*Figure 2A, B*, *Supplementary file 2*). An exception was a strong significant positive association of advancing age with CDCP1, CCL11, and CCL25, which was also observed in the Dutch cohort. Overall, these results show that the association between advancing age and inflammatory markers is much weaker in Tanzanians.

Females in both cohorts overall had lower concentrations of inflammatory proteins than males (*Figure 2A*), which is consistent with our earlier findings that females had lower ex vivo cytokine responses (*Temba et al., 2021*; *Ter Horst et al., 2016*). Results also showed that Tanzanian females had significantly higher concentrations of TNF-beta (TNFB), IL12-beta and CCL28 than Tanzanian males.

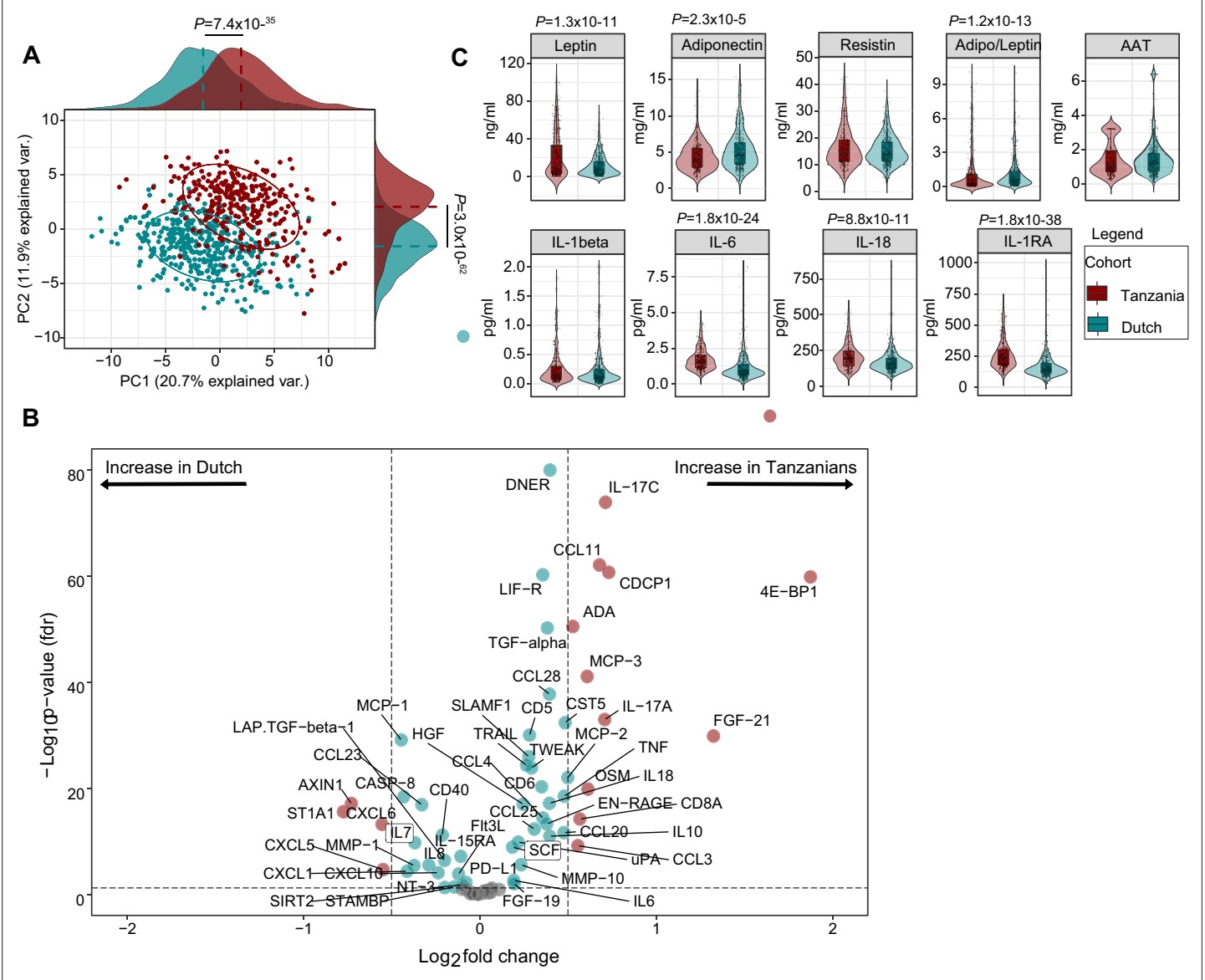

**Figure 1.** Differentially expressed inflammatory protein profiles among Dutch and Tanzanian participants. (**A**) Principal component analysis depicting the sample distribution of Dutch (N=416) vs. Tanzanian (N=318) healthy individuals across PC1 and PC2, indicating significant differences in the inflammatory protein profiles of the two cohorts. (**B**) Volcano plot showing differentially expressed proteins (DEPs) between the Dutch and Tanzanian cohorts (Dutch cohort; N=74 and Tanzanian cohort; N=72 inflammatory proteins; analyzed by Limma, linear models for microarray data, R package). The x-axis shows the Log2 fold change (Log2 FC) of the normalized protein expression (NPX), while the y-axis shows the -Log10 of the adjusted p-values (FDR <0.05); dotted lines represent the cut-off value Log2FC < 0.5 and> 0.5. (**C**) Violin plots showing concentrations of circulating adipokines and inflammatory cytokines in the Dutch and Tanzanian participants (data previously reported (*Temba et al., 2022*); differences analyzed by linear regression with age, sex and BMI as covariates). Results were declared significant after correcting for multiple testing using False discovered rate (FDR). AAT; alpha-1 antitrypsin; BMI; Body Mass Index.

The online version of this article includes the following figure supplement(s) for figure 1:

**Figure supplement 1.** Histograms and pie charts showing the distribution of cohorts characteristics comparing Tanzanian and Dutch samples.

**Figure supplement 2.** Schematic diagram showing the sample pre-processing according to the measured inflammatory protein in both cohorts.

**Figure supplement 3.** Scatter plot showing the association of plasma concentration of leptin with BMI in Tanzanian females compared to Dutch females.

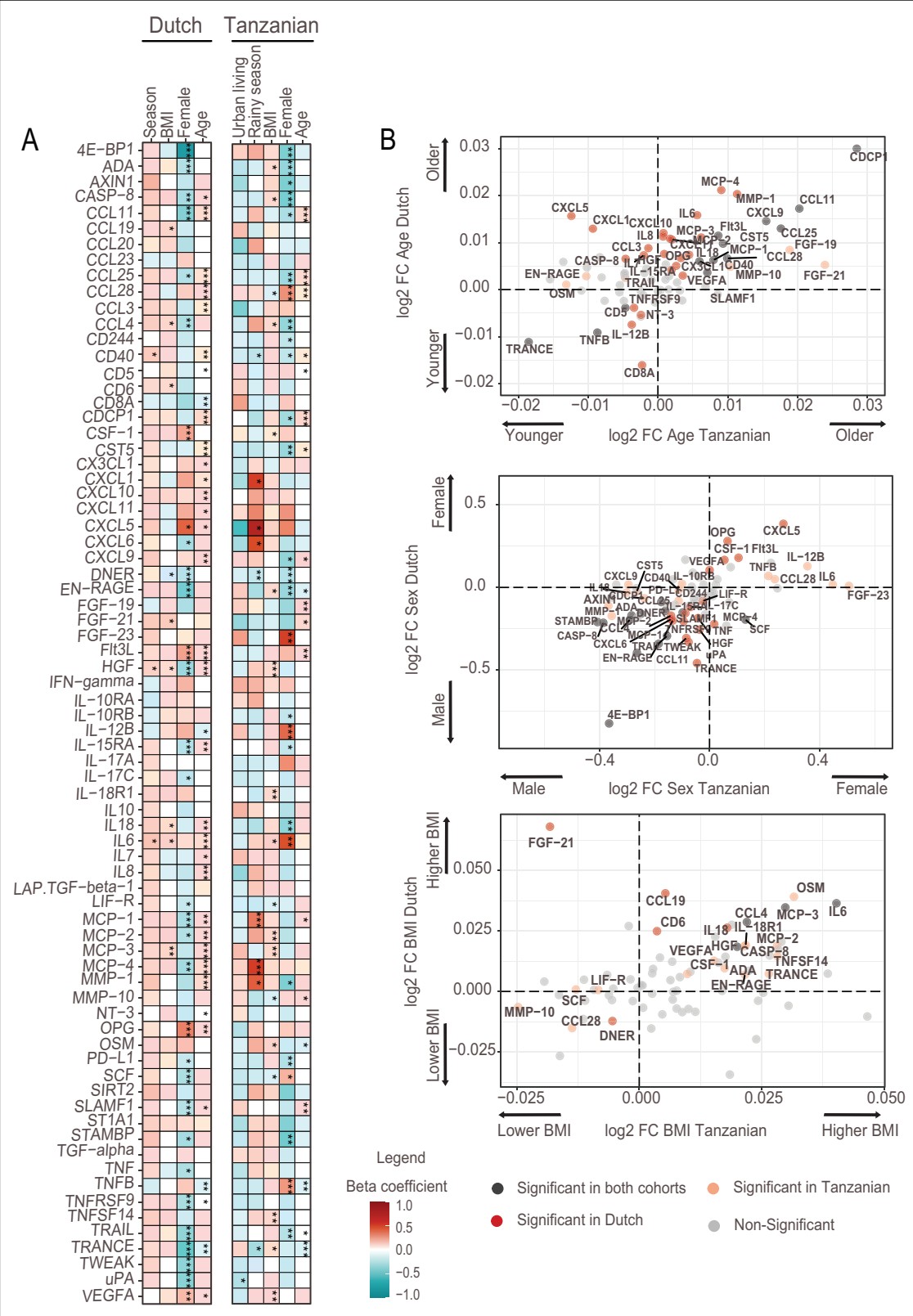

**Figure 2.** Associations of age, BMI, sex and seasonality with inflammatory proteins in Dutch and Tanzanian participants. (**A**) Heat map illustrating the regression beta-coefficient of plasma inflammatory proteins with host and environmental factors in the Dutch and Tanzanian cohorts. Red and turquoise indicate higher and lower concentrations of inflammatory proteins associated with seasonality, higher BMI, female sex, and advanced age in the Dutch cohort, and with urban living, rainy season, higher BMI, female sex, and advanced age in the Tanzanian cohort, respectively. p-Values of the significant

*Figure 2 continued on next page*

Figure 2 continued

associations are depicted, and the results were declared significant after correcting for multiple testing using the False discovered rate (FDR); p-value <0.05(*), <0.005(**), and <0.0001(***). (**B**) Four-quadrant plot depicting the association between the inflammatory protein expression with either age, sex, or BMI in the Dutch and Tanzania cohorts related to panel A.

Finally, since environmental exposures are potential drivers of inflammation, we determined the relationship between relevant exposures for Tanzanians and the proteome. Such exposures included type of toilet, exposure to wood smoke for cooking, farm-animal exposure, access to clean water, previous infections and prior use of antibiotics (*Table 1*). Our results did not show a significant association between inflammatory proteins with one of these exposures.

## Associations with food-derived metabolites and dietary habits

We recently reported that diet, and especially the transition between a rural traditional diet to an urban Western-type diet, had a major influence on ex vivo cytokine immune responses in the Tanzanian cohort (*Temba et al., 2021*). We postulated that diet also explained part of the variation in inflammatory proteins. To test this hypothesis, we selected food-derived metabolites (n=288) from an untargeted plasma metabolome, as previously described (*Temba et al., 2021*). Using these metabolites, we first performed unsupervised hierarchical clustering, which yielded two different clusters (food-metabolome clusters one and two) (*Figure 3—figure supplement 1A*). Weekly food consumption was associated with these food-metabolome clusters: participants in cluster one more frequently consumed ugali (a traditional porridge made from maize), plantain (cooking banana) and green vegetables, and less frequently rice and fried potato chips (*Figure 3—figure supplement 1B*). Next, we performed unsupervised clustering of the inflammatory proteome with age, sex, BMI, geolocation (i.e. rural vs. urban living), seasonality and the food-metabolome clusters as input variables. This analysis revealed two significant inflammatory proteome clusters: one with lower and one with higher expressed inflammatory proteins (*Figure 3*). Participants belonging to food-metabolome cluster one (i.e. more 'traditional' Tanzanian diet) were overrepresented in the cluster with lower-expressed inflammatory proteins, whereas participants belonging to cluster two were overrepresented in the cluster of higher-expressed inflammatory proteins (*Supplementary file 3*). Other factors such as age, sex or season were not associated with the inflammatory clustering.

Next, we performed a correlation analysis to assess the relationship between diet-related metabolites and inflammatory proteins. Results show different negative associations between plant-derived polyphenols (apigenin, naringenin, cyanidin 3-(6-caffeoyl glucoside) 5-glucoside, licoagrodin, shoyu-flavone C and phenolic acids such as gallic acid) and inflammatory proteins, particularly the MCP and CXCL families (*Figure 4*). In contrast, positive associations were observed between inflammatory proteins, and especially members of the MCP and CXCL families, with plasma metabolites belonging to the following classes: carboxylic acids and derivatives (e.g. aminobutanoic acid-ABA), organooxygen compounds (for example, triose), and prenol lipids such as resveratrol 4"-(6-) galloylglucoside (*Figure 4*, *Supplementary file 4*). The detailed classifications of various diet-derived metabolites and their correlations with various inflammatory proteins are presented in *Supplementary file 4*. Overall, these findings support the notion that a traditional plant-based Tanzanian diet in healthy Tanzanians has an important impact on circulating inflammatory proteins.

To confirm the impact of diet on the mTORC and Wnt/β-catenin pathways in Tanzanians, we utilized preliminary data from a proof-of-concept dietary intervention study that was conducted in the same region in Tanzania (ISCRTN15619939). To validate the importance of these pathways, we analyzed data from 23 young, healthy males residing in an urban area who underwent a 2-week dietary switch from a westernized to a traditional Tanzanian, high-fiber, plant-based diet. Using the same Olink proteomics platform, we observed a significant reduction in plasma levels of proteins associated with the mTOR pathway, including 4E-BP1 (beta coeff. −1.87; p=0.003) and FGF-19 (beta coeff. −1.253; p=0.0001). Additionally, we noticed a trend towards a reduction in the Wnt/β-catenin signaling protein AXIN1 (beta coeff. −0.555; p=0.063), while CDCP1 levels remained unchanged (beta coeff. 0.049; p=0.464).

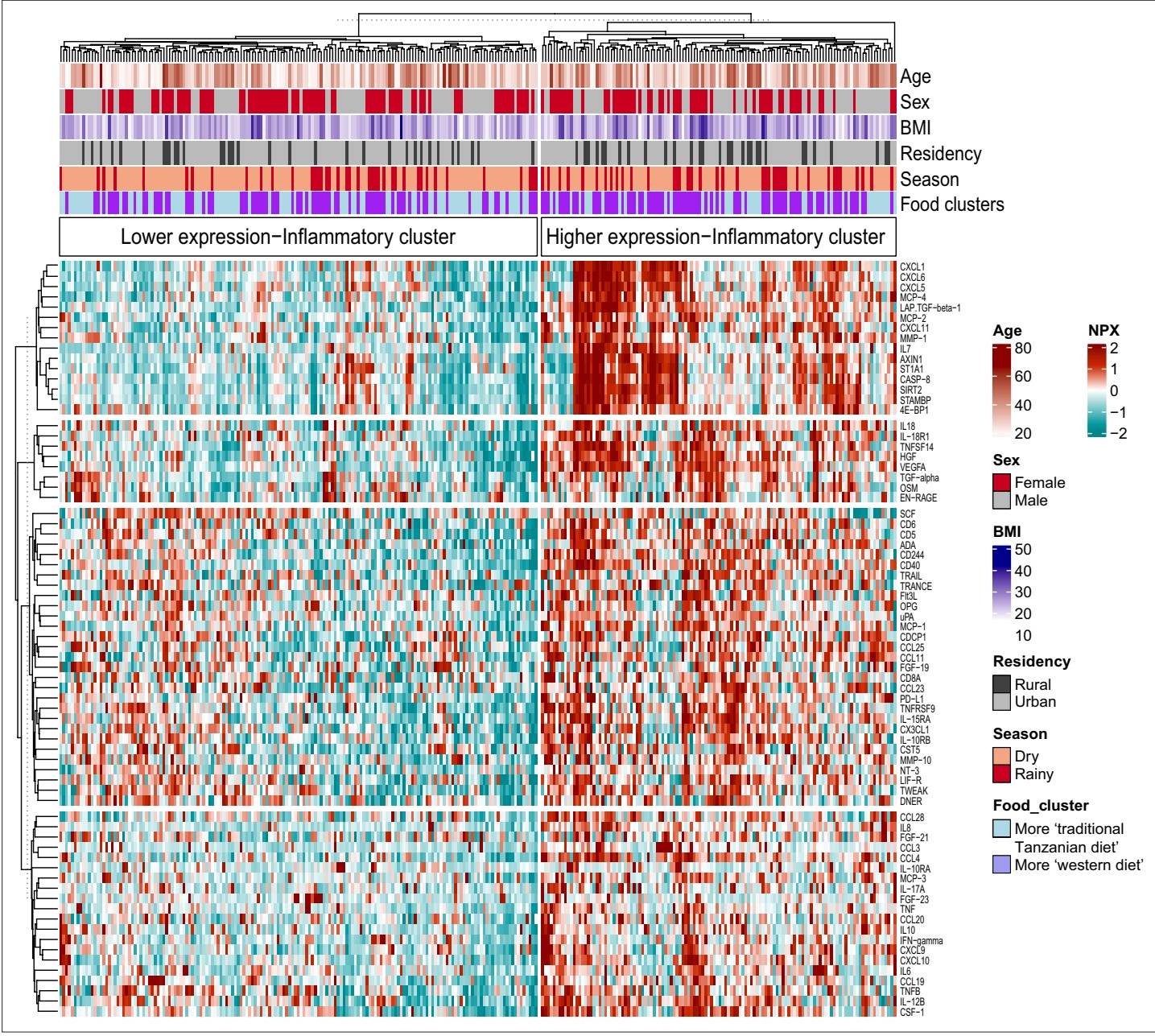

**Figure 3.** Associations between food-derived plasma metabolites and inflammation-associated proteins. Unsupervised k-means clustering of individuals from the Tanzanian cohort (N=318) according to the inflammatory proteins (N=72 inflammatory proteins). Data are shown as normalized protein expression (NPX). The color code indicates the relative expression of the inflammatory protein across the samples of the two compared groups. Dark red and turquoise colors indicate higher and lower expression, respectively. Presented are annotations for age, sex, BMI, seasonality, geographical location (i.e. rural vs. urban) and food-derived metabolite cluster. Abbreviations: NXP; normalized protein expression; BMI; body mass index.

The online version of this article includes the following figure supplement(s) for figure 3:

**Figure supplement 1.** Food-metabolome clusters and their associations with weekly food consumption within the Tanzanian cohort.

## Discussion

Genetic and environmental variations, including diet, lifestyle and infectious diseases burden, are important for modulating immune responses and may result in differences in immune phenotypes across populations. Our analysis of the inflammatory proteome shows that healthy Tanzanians have a more prominent pro-inflammatory phenotype compared to healthy Dutch individuals. Among

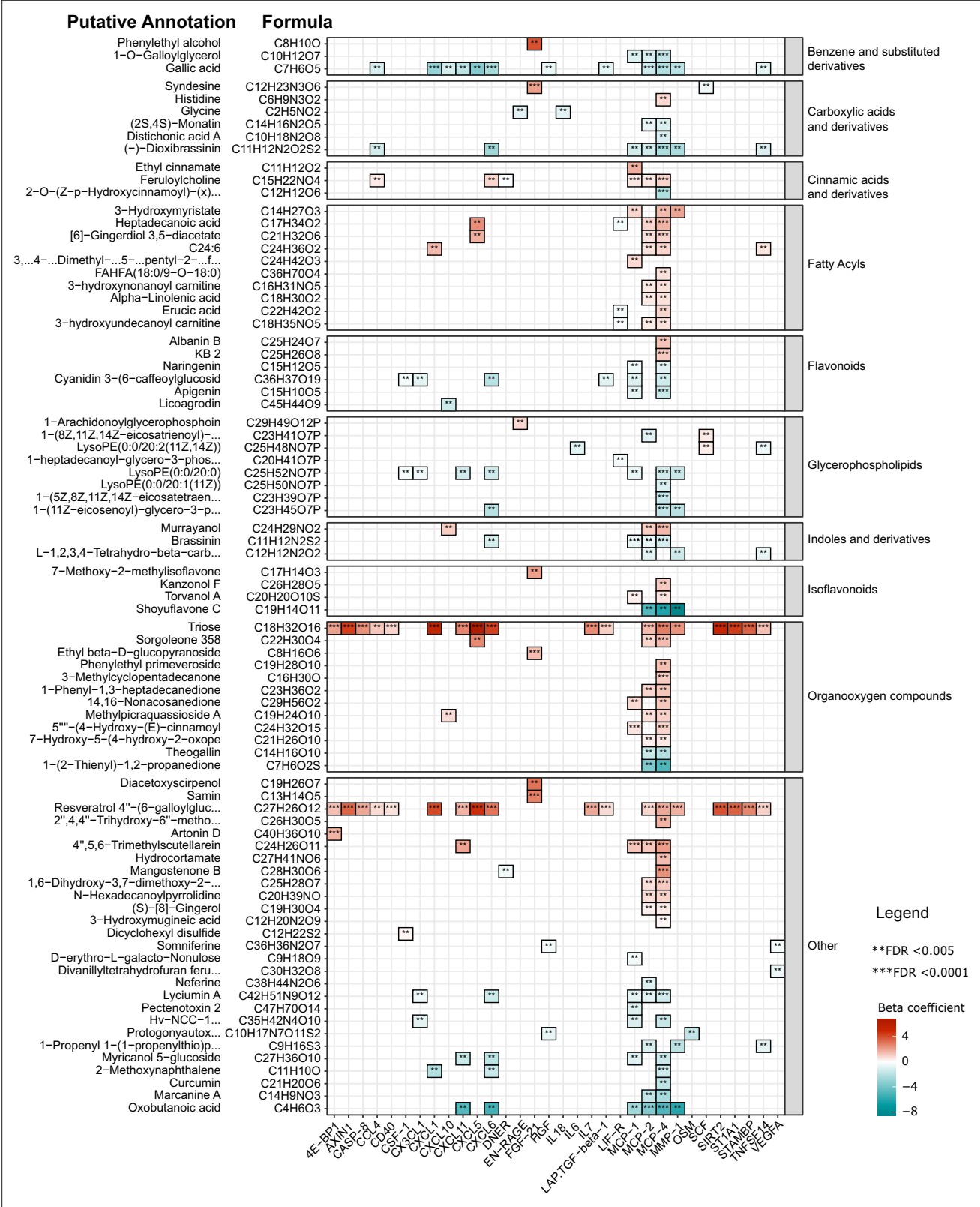

**Figure 4.** Association between inflammatory proteins and food-derived metabolites in the Tanzanian cohort. The heat map illustrates the associations between inflammatory proteins and food-derived metabolites in Tanzanian participants. The left panel displays the formulas obtained from accurate mass and natural abundance isotopic mass spectrometry data, along with the putative annotated food-derived metabolites. The β coefficients of the multiple linear regression model, including age and sex as covariates, are shown in the heat map using red and turquoise colors to indicate positive

*Figure 4 continued on next page*

*Figure 4 continued*

and negative correlations, respectively. Significance was determined after correcting for multiple testing using False Discovery Rate (FDR); p-value <0.05(*),<0.005(**), and <0.0001(***).

the Tanzanians, food-derived metabolites were identified as an important driver of variation in inflammation-related proteins, emphasizing the potential importance of lifestyle changes.

In the same European and African cohorts, we recently showed remarkable differences in the genetic regulation of immune responses, with ancestry-specific pathways regulating induced cytokine responses and significant enrichment of the interferon pathway in the Tanzanians (*Boahen et al., 2022*). From a historical perspective, the natural selection of genotypes that mediate a strong inflammatory response offers an obvious advantage in areas with a high infectious diseases burden such as sub-Saharan Africa. In addition, epigenetic regulation as a conduit that mirrors environmental exposure (e.g. diet, lifestyle or exposure to infectious diseases) may also promote an inflammatory signature. However, while a strong inflammatory response can be advantageous in the host's response against infections, it may become disadvantageous when the environment changes with a reduction in the infectious burden and a shift toward an unhealthy lifestyle. Many areas in sub-Saharan Africa are currently witnessing such changes and this is one of the important drivers of the rapid increase in non-communicable diseases and other inflammation-associated pathologies.

A healthy diet is an important component of a healthy lifestyle. In the present study, we show that Tanzanians who consumed more traditional staple foods, such as ugali (porridge made from maize or millet) or plantain, as well as green vegetables, had lower concentrations of inflammation-related proteins. We also observed several significant associations between food-derived metabolites and inflammatory proteins. Specifically, plant-derived polyphenols were negatively associated with members of the MCP and CXCL families. Especially gallic acid, a natural phenic acid with different health benefits, which is present in mangos and other edible plants (*Kim et al., 2021*), had different negative associations with immune-derived proteins. These data confirm our earlier findings that diet has an important impact on cytokine responses (*Temba et al., 2021*) as well as plasmatic coagulation (*Temba et al., 2022*). Traditional foods, particularly plant-based diets, are increasingly recognized for their cardiovascular health benefits (*Hemler and Hu, 2019*), and our findings support the notion that promoting traditional diets may be a viable public health strategy for curbing the NCDs epidemic. Other environmental exposures may potentially confound the diet-inflammation associations, for example, individuals with a traditional diet may also use more frequently a pit latrine or smoky fuels and may have more infections, but we found no significant associations between inflammatory proteins and any of these other environmental exposures.

An interesting observation was the limited number of associations between advancing age and inflammation-related proteins in the Tanzanians. In historically wealthy countries, aging is associated with a blunted response of immune cells to stimulation, while having a chronic low-inflammation state (*Yashin and Jazwinski, 2014*; *Liberale et al., 2020*), as also observed in Dutch in this study. Cytokine responses decline in the Tanzanians with advancing age (*Temba et al., 2021*), similar to Europeans, but the reasons why systemic inflammation does not increase remain uncertain. Possible explanations are the already present inflammatory signature at a younger age, possibly because of higher cumulative exposure to infections or other environmental insults, or that older participants in Tanzania manage to control the inflammatory state by a healthier lifestyle.

The two most upregulated proteins in the Tanzanians, 4E-BP1 and FGF21, are regulators of metabolism. FGF21 has received much attention recently as an endocrine regulator of glucose, lipid, and energy metabolism (*Hill et al., 2018*). FGF21 is an insulin sensitizer, and its secretion is increased by a high intake of carbohydrates and low dietary protein intake (*Maekawa et al., 2017*; *Lundsgaard et al., 2017*). FGF21 inhibits age-associated metabolic syndrome and protects against diabetic cardiomyopathy (*Yan et al., 2021*). FGF21, therefore, acts differently from classic energy balance signals like leptin (*Hill et al., 2018*). Traditional Tanzanian diets are high in carbohydrates and low in proteins, which may explain the high FGF21 among the Tanzanians. FGF21 was also reported to be higher in obesity (*Zhang et al., 2008*), but we did not find an association with BMI. Tanzanians also had higher leptin and lower adiponectin concentrations than the Dutch. This is in line with the results of an earlier study among non-obese and obese subjects with type 2 diabetes in Tanzania and Sweden, which showed that leptin concentrations were 50% higher in the Tanzanians (*Abbas et al., 2004*).

This supports the importance of potential ethnic differences in metabolic profiles and adipocytokines (**Mente et al., 2010**). To our knowledge, this study is the first to compare FGF21 levels between healthy individuals from sub-Saharan Africa and Europe.

Another interesting finding was the significant differences in CDCP1 and AXIN1, pointing towards an enhanced activity of the Wnt/β-catenin pathway in the Tanzanians. The Wnt/β-catenin pathway is a key regulator of inflammation, playing a role in both the inflammatory and anti-inflammatory pathways (**Ma and Hottiger, 2016**). Dysregulated activation of the Wnt/β-catenin pathway is increasingly recognized to play a role in the pathogenesis of chronic inflammatory diseases, metabolic inflammatory diseases and cancer (**Jridi et al., 2020**). Mice fed on a high-fat Western-type diet expressed high concentrations of Wnt2 protein in atherosclerotic lesions, suggesting that the Wnt/β-catenin pathway also contributes to atherosclerosis (**Zhang et al., 2021**).

The resolution of inflammation is typically coordinated by key lipids, proteins, and peptides (such as annexin, lipoxins, and resolvins) (**Headland and Norling, 2015**). It would be interesting to examine whether a decrease in the production of these endogenous mediators by immune cells could potentially contribute to the pro-inflammatory phenotype observed in the Tanzanians.

To summarize, our findings reveal significant differences in inflammatory and metabolic proteins and pathways between healthy individuals living in East Africa and individuals living in Western Europe. This is especially important in light of the current epidemiological transition and lifestyle changes in sub-Saharan Africa, which coincides with a sharp increase in non-communicable diseases in the region. Our study also endorses the importance of including underrepresented populations in systems-immunology studies.

# Materials and methods

## Study design and population

The present study used samples from two cross-sectional cohorts of healthy volunteers: the 300-Tanzania-FG (300TZFG) and the Dutch 500FG. Both cohorts were enrolled within the Human Functional Genomics Project (https://www.humanfunctionalgenomics.org). The demographic characteristics of both cohorts have been described previously (**Temba et al., 2021**; **Ter Horst et al., 2016**). Briefly, the 300TZFG cohort consists of 323 healthy Tanzanian individuals aged between 18 and 65 years residing in the Kilimanjaro region in Northern Tanzania. The cohort was enrolled between March and December 2017. Exclusion criteria were participants with any acute or chronic disease, use of antibiotics or anti-malaria medication in the three months before blood sampling, tuberculosis in the past year, a blood pressure ≤90/60 mmHg or ≥140/90 mmHg, or random blood glucose >8.0 mmol/L. Pregnant, postpartum, or breastfeeding females were excluded. The 500FG cohort consists of 534 Dutch individuals of Western-European background, aged 18 years and older. Data was collected between August 2013 and December 2014 at the Radboud university medical center (Radboudumc) in the Netherlands. Exclusion criteria were: the use of any medication in the past month and acute or chronic diseases at the time of blood sampling. Pregnant, postpartum, or breastfeeding females were excluded.

## Sample collection and preparation

The current study is part of the Human Functional Genomics Project (HFGP; humanfunctionalgenomics.org), which employs standardized procedures for sample collection, handling, and pre-processing. Blood was obtained in the morning via antecubital puncture into ethylenediaminetetraacetic acid (EDTA) tubes (Monoject; Covidien, Ireland). Within two to three hours after blood collection, plasma was collected by centrifugation at 3800 rpm for 8 min at room temperature. The obtained plasma were stored at −80 °C, as recommended by the ISBER biobanking organization (**Garcia et al., 2014**). Plasma samples for the Tanzania cohort were shipped to the Netherlands on dry ice.

## Inflammatory proteome

Plasma proteins were measured with the Olink 92 Inflammation panel using proximity extension technology (Olink Proteomics AB, Uppsala, Sweden) (**Assarsson et al., 2014**). This panel includes 92 inflammation-related proteins. This assay utilizes the binding of target proteins by paired oligonucleotide antibody probes, followed by hybridization and amplification. Data are reported as normalized

protein expression values (NPX), which is an arbitrary unit in a Log2 scale that is calculated from normalized Ct values. Validation data of the assay are available on the Olink website (https://www.olink.com). All samples were measured in the same batch in a single run. Proteins were excluded from analysis when values were both below the detection limit in more than 25% of all samples. Plasma samples from the Tanzanian and Dutch cohorts were on their first and second freeze-thawed cycles, respectively. Pre-analytical processing such as freeze-thawed cycles and storage time has limited influence on the measured proteins reported in this study (*Shen et al., 2018*; *Enroth et al., 2016*; *Lee et al., 2015*).

## Measurement of the circulating inflammatory mediators

Plasma concentrations of the cytokines IL-6, IL-1β, IL-1 receptor antagonist (IL-1Ra) and IL-18 (lot number Bio-Tech/R&D; SPCKC-PS-001559) and IL-18 binding protein (IL-18BPa) (lot number Bio-Tech/R&D; SPCKB-PS-000502) were measured in EDTA plasma using the Simple Plex cartridges run on the Ella platform (Protein Simple, San Jose, USA) following the manufacturer's instructions.

## Plasma metabolome

Plasma samples of the Tanzanian cohort were measured using the untargeted metabolomics workflow by General Metabolics (Boston, MA) with procedures as previously described (*Fuhrer et al., 2011*). In short, metabolites were measured by a high throughput mass spectrometry technique using the Agilent Series 1100 LC pump coupled to a Gerstel MPS2 autosampler and the Agilent 6520 Series Quadrupole Time-of-flight mass spectrometer (Agilent, Santa Clara, CA). Our non-targeted high-throughput method relies on flow injection and does not involve separation of compounds based on an LC gradient. This is because many different species would be fragmented simultaneously. Historically, the decision to collect high-accuracy flow injection has been a technological trade-off to enable efficient analysis of large cohorts. Therefore, metabolites detected in the high-throughput non-targeted metabolomics screening method, which enables the screening of these large cohorts for metabolomic feature patterns, are based on accurate mass (approximately 1 ppm accuracy) and natural isotope detection consistent with the assigned formulae; final assignments are pending validation with LC-MS/MS. For this study, the MS spectra files (accessible at http://www.ebi.ac.uk/metaboLights/MTBLS2267) contain 70 scans, each of which contains the full MS1 profile data from m/z 50–1050 for two subsequent injections from the same needle draw of any given sample. The analytical method and approach to data processing and annotation were previously described (*Fuhrer et al., 2011*). The selection of food-derived metabolites was performed based on the ontology given in the HMDB (https://www.hmdb.ca/) as described previously (*Temba et al., 2021*).

## Statistical analysis

The proteomic data from the Dutch and Tanzanian cohorts were normalized using inter-plate controls for batch variation correction and presented in the $\log_2$ scale. Normalizing for batch effects was done using pooled plasma standards on all plates. Eight samples of plasma pool controls were measured in each cohort (two per plate, a total of 4 plates per cohort). We performed bridging normalization based on median differences between plasma pool controls in the 300TZFG and 500FG cohorts. The following steps were applied for each protein: (1) Determine the median value from the eight bridging samples for each protein in the 300TZFG and 500FG cohort. (2) Calculate the median differences (300TZFG-500FG) as median differences X. (3) Take the NXP values of each protein in the 300TZFG cohort and subtract the median difference (the X value). This created a normalized data set for the 300TZFG cohort.

Data values below the limit of detection (LOD) were handled using the actual measured values to increase the statistical power and give a complete data distribution. Outliers detection were done using principal component analysis (PCA) in which data points that fall in more than 3 standard deviations from the mean of principal component one (PC1) and two (PC2) were excluded. This pre-analytical process led to exclusion of 12 participants (N=7 Dutch and N=5 Tanzanians) as potential outliers. Proteins with >25% data values below LOD in both cohorts were excluded (N=18 in the Duch and N=20 in the Tanzania cohorts), leaving 74 and 72 inflammatory proteins for the downstream data analysis in the Dutch and Tanzanian cohorts respectively. In total, 416 Dutch and 318 Tanzania

participants were available after the pre-analytical process. Details of preanalytical steps for both cohorts are described in *Figure 1—figure supplement 2*.

To analyze the similarities and dissimilarities between the samples, unsupervised PCA was performed using 'prcomp' function in R package. Heatmap of unsupervised hierarchical clustering (k-nearest neighbors with 100 repetitions) of the samples was generated using 'ComplexHeatmap' R package by calculating the matrix of Euclidean distances from the $log_2$ NPX value. Limma (linear models for microarray data) R package was used for differential expression analysis of plasma inflammatory proteins between cohorts. Proteins with adjusted P-value (FDR)<0.05 were selected as significantly differentially expressed (DE). Proteins with a positive and negative value of $log_2$-fold-change were considered as upregulated and downregulated, respectively.

## Acknowledgements

The authors wish to express their gratitude to all volunteers in the Human Functional Genomics in Tanzania and Dutch cohorts for their participation. We would like to thank J Njau, J Kwayu, and E Kimaro for assistance with sample collection, H Lemmers, and H Toenhake-Dijkstra for assistance with laboratory analysis, and M Miclaus for assistance with pre-processing metabolome data. This study was funded by the following grants: the European Union's Horizon 2020 Research and Innovation Program under the ERA-Net Cofund action no. 727565; the Joint Programming Initiative, A Healthy Diet for a Healthy Life (JPI-HDHL); The Netherlands Organization for Health Research and Development (ZonMW, grant number 529051018; TransMic) awarded to MGN, QdM, DC, PL and JLS; ZonMw (the Netherlands Organization for Health Research and Development) awarded to MGN, QdM and AV; Radboud Revolving Research Funds (3R-Fund) awarded to GST; Indonesia Endowment Fund for Education (LPDP) given by the Ministry of Finance of the Republic of Indonesia awarded to NV; Spinoza Prize (NWO SPI94-212) and ERC Advanced grant (no. 833247) awarded to MGN; and the Deutsche Forschungsgemeinschaft (German Research Foundation) under Germany's Excellence Strategy (EXC2151) 390873048 awarded to MGN and JLS.

## Additional information

### Funding

| Funder | Grant reference number | Author |
|---|---|---|
| Joint Programming Initiative A healthy diet for a healthy life | TransMic (project 529051018) | Mihai G Netea<br>Quirijn de Mast<br>Duccio Cavalieri<br>Paolo Lionetti<br>Joachim L Schultze |
| ZonMw | | Mihai G Netea<br>Quirijn de Mast<br>Andre J van der Ven |
| Radboud Revolving Research Funds | | Godfrey S Temba |
| Ministry of Finance of Republic of Indonesia | Indonesia Endowment Fund for Education | Nadira Vadaq |
| Netherlands Organisation for Scientific Research | Spinoza Prize NWO SPI94-212 | Mihai G Netea |
| European Research Council | Advanced grant 833247 | Mihai G Netea |
| Deutsche Forschungsgemeinschaft | EXC2151 390873048 | Mihai G Netea<br>Joachim L Schultze |

The funders had no role in study design, data collection and interpretation, or the decision to submit the work for publication.

## Author contributions
Godfrey S Temba, Conceptualization, Data curation, Investigation, Methodology, Writing - original draft; Nadira Vadaq, Data curation, Formal analysis, Writing - review and editing; Vesla Kullaya, Investigation, Methodology, Writing - review and editing; Tal Pecht, Data curation, Formal analysis, Methodology, Writing - review and editing; Paolo Lionetti, Supervision, Funding acquisition, Writing - review and editing; Duccio Cavalieri, Conceptualization, Supervision, Funding acquisition, Writing - review and editing; Joachim L Schultze, Mihai G Netea, Conceptualization, Supervision, Funding acquisition, Methodology, Writing - review and editing; Reginald Kavishe, Conceptualization, Supervision, Project administration, Writing - review and editing; Leo AB Joosten, Conceptualization, Supervision, Methodology, Writing - review and editing; Andre J van der Ven, Conceptualization, Supervision, Writing - review and editing; Blandina T Mmbaga, Supervision, Project administration, Writing - review and editing; Quirijn de Mast, Conceptualization, Supervision, Funding acquisition, Methodology, Writing - original draft, Project administration, Writing - review and editing

## Author ORCIDs
Godfrey S Temba (iD) http://orcid.org/0000-0002-1093-3037
Vesla Kullaya (iD) http://orcid.org/0000-0001-6120-3985
Leo AB Joosten (iD) http://orcid.org/0000-0001-6166-9830
Andre J van der Ven (iD) http://orcid.org/0000-0003-1833-3391
Mihai G Netea (iD) http://orcid.org/0000-0003-2421-6052
Quirijn de Mast (iD) http://orcid.org/0000-0001-6056-157X

## Ethics
Ethical statementThe study was approved by the Ethical Committee of the Kilimanjaro Christian Medical University College (CRERC) (No 2443) and the National Institute for Medical Research (NIMR/HQ/R.8a/Vol. IX/2290 and NIMR/HQ/R.8a/Vol.IX/3318) in Tanzania. The 500FG cohort study was approved by the Ethical Committee of the Radboud University Medical Centre Nijmegen, the Netherlands (NL42561.091.12, 2012/550). Subject recruitment and experimental procedures were conducted according to the principles mentioned in the Declaration of Helsinki. Written informed consent was obtained from all subjects.

## Decision letter and Author response
Decision letter https://doi.org/10.7554/eLife.82297.sa1
Author response https://doi.org/10.7554/eLife.82297.sa2

---

# Additional files

## Supplementary files
• Supplementary file 1. The table shows the top SNPs that were independently associated with ex vivo cytokine (*Boahen et al., 2022*) and its correlation with the 74 inflammatory proteins measured in the same cohort. P- value unadjusted.

• Supplementary file 2. The table summarizes the analyzed data on the differentially expressed inflammatory protein among Dutch and Tanzanian participants. This table relates to the data displayed in *Figure 1*.

• Supplementary file 3. The table shows the analyzed data on the associations between food-derived plasma metabolites and inflammation-associated proteins. This table is related to the data displayed in *Figure 3*.

• Supplementary file 4. The table displays data from the multivariate linear regression analysis between metabolites and protein expression concentrations in the Tanzanian cohort. This table is related to the data presented in *Figure 4*.

• MDAR checklist

• Source data 1. Normalized protein expression (NPX) units (log2) of the inflammatory proteins in the Tanzanian and Dutch participants.

## Data availability
Anonymized metadata of the Tanzanian participants and the circulating inflammation markers are available in an open access registry (DANS registry; https://doi.org/10.17026/dans-xgx-zuht). Untargeted

plasma metabolome data have been deposited to the EMBL-EBI MetaboLights database (http://www.ebi.ac.uk/metabolights/) with study identifier MTBLS2267. The source data of the proteomics analysis are provided in *Source data 1*. Publicly available databases used for this study include KEGG (https://www.genome.jp/kegg/), HMDB (https://www.hmdb.ca/) and ChEBI (https://ebi.ac.uk/chebi/). All other data is available in the main text and supplementary materials.

The following previously published datasets were used:

| Author(s) | Year | Dataset title | Dataset URL | Database and Identifier |
|---|---|---|---|---|
| de Mast Q, Temba GS, Kullaya V, Pecht T, Mmbaga BT, Aschenbrenner AC, Ulas T, Kibiki G, Lyamuya F, Boahen CK, Kumar V | 2021 | Urban living in healthy Tanzanians is associated with an inflammatory status driven by dietary and metabolic changes | https://www.ebi.ac.uk/metabolights/editor/MTBLS2267 | EBL MetaboLights, MTBLS2267 |
| de Mast Q, Temba GS | 2022 | Human functional genomics in healthy Tanzania individuals: A system biology approach in understanding individual variations in immune response | https://doi.org/10.17026/dans-xgx-zuht | DANS, 10.17026/dans-xgx-zuht |

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
