## [Editor Report]

The manuscript by Temba and colleagues describe an essential aspect of human research, i.e., the variability of different populations which have different genetic backgrounds and are exposed to different diets and environments. As most research is carried out in developed countries, we need to understand and generate background data in other populations, such as the described here in Tanzania. The fundamental findings refer to enhanced pro-inflammatory phenotype in plasma of individuals from Tanzania as compared to individuals from Holland. Food-derived metabolites were identified as a driver of variation in inflammation-related molecule expression. Whether pro-inflammatory phenotypes associate with changes in life span or risk of chronic disease and whether changes in life style will reverse this pro-inflammatory phenotype and related outcomes clearly deserve further investigation in the future.

---

## [Decision Letter]

**Decision letter after peer review:**

Thank you for submitting your article "Differences in the inflammatory proteome of East African and Western European adults and associations with environmental and dietary factors" for consideration by *eLife*. Your article has been reviewed by 2 peer reviewers, and the evaluation has been overseen by a Reviewing Editor and Carlos Isales as the Senior Editor. The reviewers have opted to remain anonymous.

Essential revisions:

1) Both reviewers and I found the manuscript interesting and important. Both reviewers would like to see some validation of findings – either ex vivo, in vivo or in different populations. This may be a little beyond the scope of the current study (as it would need large patient cohorts or significant clinical studies) but needs to be discussed in some detail. Whenever possible, validation should be performed, as suggested by reviewers.

2) There is a need to clarify some relevant methodological issues.

*Reviewer #1 (Recommendations for the authors):*

Recommendations for the authors:

– The proteomic data were normalized for batch effect. The authors should describe in detail the batch correction method.

– The authors should correlate the inflammatory proteome with the common genetic variations described in a previous manuscript for the same cohorts (Boahen et al., 2022).

– Circulating proteins related to the mTOR pathway and metabolism, such as 4EBP1 and FGF21, were differentially regulated. Moreover, AXIN1 and CDCP1, associated with the Wnt/β-catenin pathway, were regulated. The authors should validate the differential expression of these pathways by looking at other targets in the two cohorts using orthogonal techniques.

–The dysregulation of mTOR and Wnt/β-catenin pathways should be evaluated upon food-borne metabolite treatment. This validation can be performed in ex vivo and in vivo models.

– From the Materials and methods section, the identification of metabolites was based on accurate mass. However, no fragmentation spectra were shown to validate the identity of the metabolites. Fragmentation spectra should be shown to validate the identifications of food-borne metabolites.

*Reviewer #2 (Recommendations for the authors):*

Overall, this is an interesting study that has potential broad impact if the findings can be reproduced in additional cohorts. I have no major concerns. The authors are correct to state that genetics and epigenetic mechanisms may play are strong role here with respect to population differences and impact on inflammatory proteins and specific metabolites.

It would be interesting to confirm these findings using additional sub-Saharan Africa cohorts in a more targeted way – for example, could lead metabolites and cytokines be measured in additional cohorts from sub-Saharan Africa?

An alternative explanation to consider here is that there is an impairment in the resolution of inflammation in the Tanzanian cohort. Resolution of inflammation is typically coordinated by key lipids, proteins and peptides (E.g. annexin, lipoxins, resolvins etc…). Could it be that there is impaired production of these endogenous mediators by immune cells in the Tanzanian cohort versus the Dutch cohort? This is speculative, but it could be mentioned as a possibility for future studies.

The selected o-link panel obviously biases the analyses towards inflammatory pathways, and it would be interesting in future to consider using a more global proteomics approach to identify alternative pathways that may be contributing factors here.

---

## [Author Response]

Essential revisions:Reviewer #1 (Recommendations for the authors):Recommendations for the authors:– The proteomic data were normalized for batch effect. The authors should describe in detail the batch correction method.

We now describe the normalization for batch effect in detail in the methods section under statistical analysis, page 20; lines 320-327. Normalizing for batch effects was done using pooled plasma standards on all plates. Eight samples of plasma pool controls were measured in each cohort (two per plate, a total of 4 plates per cohort). We performed bridging normalization based on median differences between plasma pool controls in the 300TZFG and 500FG cohorts. The following steps were applied for each protein: (1) Determine the median value from the eight bridging samples for each protein in the 300TZFG and 500FG cohort. (2) Calculate the median differences (300TZFG-500FG) as median differences X. (3) Take the NXP values of each protein in the 300TZFG cohort and subtract the median difference (the X value). This created a normalized data set for the 300TZFG cohort.

– The authors should correlate the inflammatory proteome with the common genetic variations described in a previous manuscript for the same cohorts (Boahen et al., 2022).

We thank the reviewer for this suggestion. In our previous manuscript (Boahen et al., 2022), we studied the genetic variations of whole-blood cytokine responses to a range of microbial and synthetic ligands. We used the top six independent SNP-cytokine loci identified in this previous paper and performed matrix pQTL analysis to determine an association with the protein expression levels. As shown in the table, we found several associations, but none of these reached genome-wide (P<5X10^-8^) or suggestive (P<5X10^-6^) significance, possibly due to the limited sample size of our cohort.

We have included this information in the text (page 9; lines 107-112) and we included the table as a Table S1.

– Circulating proteins related to the mTOR pathway and metabolism, such as 4EBP1 and FGF21, were differentially regulated. Moreover, AXIN1 and CDCP1, associated with the Wnt/β-catenin pathway, were regulated. The authors should validate the differential expression of these pathways by looking at other targets in the two cohorts using orthogonal techniques.

We attempted to validate the differential expression of these proteins and pathways using whole blood transcriptomics data, but we encountered challenges with comparability between the cohorts. Therefore, we were unable to validate the differences between the Tanzanian and Dutch cohorts using orthogonal techniques. However, we were able to confirm the significance of these pathways in relation to traditional versus westernized diet, as described in the following comment.

–The dysregulation of mTOR and Wnt/β-catenin pathways should be evaluated upon food-borne metabolite treatment. This validation can be performed in ex vivo and in vivo models.

In response to the comment, we utilized preliminary data from a proof-of-concept dietary intervention study that was conducted in the same region in Tanzania (ISCRTN15619939). To validate the importance of these pathways, we analyzed data from 23 young, healthy males residing in an urban area who underwent a 2-week dietary switch from a westernized to a traditional Tanzanian, high-fiber, plant-based diet. Using the same Olink proteomics platform, we observed a significant reduction in plasma levels of proteins associated with the mTOR pathway, including 4E-BP1 (β coeff. -1.87; *P* = 0.003) and FGF-19 (β coeff. -1.253; *P* = 0.0001). Additionally, we noticed a trend towards a reduction in the Wnt/ β -catenin signaling protein AXIN1 (β coeff. -0.555; *P* = 0.063), while CDCP1 levels remained unchanged (β coeff. 0.049; *P* = 0.464). We have included these findings on page 12; lines 166-175.

– From the Materials and methods section, the identification of metabolites was based on accurate mass. However, no fragmentation spectra were shown to validate the identity of the metabolites. Fragmentation spectra should be shown to validate the identifications of food-borne metabolites.

We appreciate the reviewer's suggestion and have now provided a detailed description of the metabolite measurement method in the methods section. Our non-targeted high-throughput method relies on flow injection and does not involve separation of compounds based on an LC gradient. This is because many different species would be fragmented simultaneously. Historically, the decision to collect high-accuracy flow injection has been a technological trade-off to enable efficient analysis of large cohorts. Therefore, metabolites detected in the high-throughput non-targeted metabolomics screening method, which enables the screening of these large cohorts for metabolomic feature patterns, are based on accurate mass (approximately 1 ppm accuracy) and natural isotope detection consistent with the assigned formulae; final assignments are pending validation with LC-MS/MS. For this study, the MS spectra files (accessible at http://www.ebi.ac.uk/metaboLights/MTBLS2267) contain 70 scans, each of which contains the full MS1 profile data from m/z 50 to 1050 for two subsequent injections from the same needle draw of any given sample.

We have incorporated this information into the method section of the revised manuscript (page 19; lines 296-308), and we have updated Figure 4 to include the putatively annotated food-borne metabolites with their formulas next to them.

Reviewer #2 (Recommendations for the authors):Overall, this is an interesting study that has potential broad impact if the findings can be reproduced in additional cohorts. I have no major concerns. The authors are correct to state that genetics and epigenetic mechanisms may play are strong role here with respect to population differences and impact on inflammatory proteins and specific metabolites.It would be interesting to confirm these findings using additional sub-Saharan Africa cohorts in a more targeted way – for example, could lead metabolites and cytokines be measured in additional cohorts from sub-Saharan Africa?

We thank the reviewer for his evaluation of the manuscript and the suggestions. Regarding the validation, we refer to our response to reviewer 1 above.

An alternative explanation to consider here is that there is an impairment in the resolution of inflammation in the Tanzanian cohort. Resolution of inflammation is typically coordinated by key lipids, proteins and peptides (E.g. annexin, lipoxins, resolvins etc…). Could it be that there is impaired production of these endogenous mediators by immune cells in the Tanzanian cohort versus the Dutch cohort? This is speculative, but it could be mentioned as a possibility for future studies.

We appreciate the reviewer's interesting suggestion. While we do possess plasma metabolome data for both cohorts, we are unable to compare them due to the fact that they were measured in different batches. We have included this interesting hypothesis in the Discussion section (page 16; lines 243-246).

The selected o-link panel obviously biases the analyses towards inflammatory pathways, and it would be interesting in future to consider using a more global proteomics approach to identify alternative pathways that may be contributing factors here.

We agree with the reviewer that our analysis is restricted to inflammation-related pathways. Further proteomic panels will undoubtedly be incorporated in subsequent studies, including the dietary-intervention study mentioned above.